# A Local Approach to Better Understand the Spread and Population Growth of the Monk Parakeet as an Invasive Species

**Sandro López-Ramírez**  **and Antonio-Román Muñoz** *

Biogeography, Diversity and Conservation Research Team, Department of Animal Biology, Faculty of Sciences, Universidad de Málaga, 29017 Malaga, Spain
* Correspondence: roman@uma.es

**Simple Summary:** The scientific community and society, in general, have a growing interest in invasive exotic species. This is due to the problems they cause or may cause once they are established in new ecosystems when their control and eradication may be difficult and costly. The aim of this study was to monitor the growth of a Monk Parakeet population at a local level to determine the maximum sustainable population size of the invaded area and to check when this population began to act as a source for the establishment of new breeding colonies in adjacent areas. The first breeding of the species in our study site was detected in 2002. The average growth rate ($r$) of the population was 0.191 until 2016 when the population stabilized and declined slightly ($r = -0.043$). The maximum growth rate was 0.314 and occurred during the years 2002 and 2007. The results suggest that there may be a limit to the number of parakeets in local populations that, once reached, may favour the dispersal of individuals to neighbouring areas. Once they have reached maximum size, or maybe earlier, populations may serve as sources of Monk Parakeets for the colonization of favourable adjacent areas. Monk Parakeets show a preference for building their nests in palms and on electric utility structures, and this preference should be considered in management actions. We conclude that it is necessary to know, in detail, the dispersal process and population change over time, bearing in mind that prevention and early intervention are always better than cure.

**Abstract:** Invasive exotic species are currently a topic of interest for environmental management agencies and the media. This is due to the socio-economic and environmental problems that they are causing or may cause. The Monk Parakeet is a clear example of this, especially in some large cities, where populations are growing quickly, and their distribution continues to expand. In our study, we focused on a population that has been closely monitored during the last 25 years to understand its growth and change on a local scale to determine the maximum sustainable population size in the invaded area and to check when this population began to act as a source of new breeding colonies to adjacent areas. The first breeding of the species occurred in 2002, and from then, the average growth rate ($r$) of the population was 0.191 until 2016, when it stabilized and declined slightly ($r = -0.043$). The maximum growth rate occurred during the years 2002 and 2007 and was 0.314. The results obtained give us information about the carrying capacity of the study area, allowing us to explain the dispersal process of the species to neighbouring areas, with populations that have reached maximum size serving as sources of individuals dispersing to suitable areas. In our study area, the species shows a strong preference for building its nests in palm trees and electric utility structures. Our study's local focus on studying the population dynamics of an invasive species may allow us to understand the increased range of the species on a larger scale, which is necessary in order to be able to design appropriate and effective management strategies.

**Keywords:** biological invasions; bird behaviour; invasive exotic species; maximum population size; *Myiopsitta monachus*; southern Spain

## 1. Introduction

Exotic species are those that have been transported out of their natural range as a direct or indirect result of human actions [1]. The successful exotic species that manage to survive are considered to be invasive if they establish themselves in a new habitat and can reproduce and spread effectively from the area where they were initially introduced. Invasive species may also pose a significant threat to the species, ecosystem, and/or economy of their new range [2,3]. However, most introduced species do not become invasive [4,5]. In southern Spain, more than 120 species of exotic birds have been recorded, but less than 10% have managed to establish self-sustaining populations. In the last century, a rise in the legal and illegal pet trade has dramatically boosted animal movements across international borders [6]. If this trade is uncontrolled, it opens the gateway for biological invasions [7]. Parrots (Psittaciformes), among the most popular birds to be traded for pets, are known to successfully become established in new environments as a result of accidental escapes or deliberate releases [8,9], as is the case of the Monk Parakeet (*Myiopsitta monachus*) in Spain [10].

The Monk Parakeet is native to South America and has been a popular pet since before the 1970s [11]. Because of that, millions of individuals were captured and exported from their native habitat to different parts of the world [12,13]. Some characteristics of the species favoured their successful establishment in new habitats. This parakeet has a loud and harsh call, which is often the reason why many owners have released these birds [12]. These deliberate releases, accompanied by accidental escapes from captivity, have caused the fast spread that nowadays involves four continents: North America, Asia, Africa and Western Europe (especially Spain, Italy and Belgium) [14–17]. Another important characteristic is that the Monk Parakeet is the only species within the Psittacidae family that builds its nests rather than using cavities for breeding [18]. This facilitates its establishment without having to compete with other species for available cavities for nesting [19,20]. In addition, it is a gregarious species that is well adapted to successfully exploit a large variety of native and introduced plants in urban environments [21].

The first record of a free-living Monk Parakeet in Spain dates back to 1975, when two individuals were observed in Barcelona and Murcia urban areas [13,22]. In Malaga, where our study area is located, the first record was detected in 1978 [23]. Since then, this species has expanded its range throughout Spain, especially in and around large cities [12,22,24–26], including Madrid [27] and Malaga [28]. Malaga city has one of the largest populations in Spain, with 2181–2467 individuals in 2015 [13]. At present, the species' distribution and population size continues to grow and has the potential for spreading even more widely [29,30]. Like many other invasive exotic species, after becoming established, the Monk Parakeet population has grown exponentially since the first breeding individuals were detected [10,13,31,32]. In Malaga and Spain, we found this situation [30,33]. However, at a more local level, populations may reach a maximum sustainable level of density, possibly indicating that the maximum level of carrying capacity has been reached. This could favour the spread of the species in the vicinity of those populations with high levels of density and could help to understand its spread.

The carrying capacity of an environment is the maximum population size of a species that can be maintained without negative effects on the habitat [34]. This maximum population size depends on available resources, such as water and food supply, available nesting substrates, as well as intra or interspecific competition [35]. Other key aspects that could play an important role in the population dynamics of the species are the survival rate and the mortality (natural or non-natural), and the effects derived from the population's density.

The present study aimed to monitor the growth of a Monk Parakeet's population at a local level starting from the time at which the species was first recorded in the area, to determine the area's carrying capacity and whether the population had become a source of new breeding colonies in adjacent areas.

## 2. Materials and Methods

### 2.1. Study Area

The study area was located in southern Spain and comprised a 58.360 m² sports facility (Ciudad Deportiva de Carranque) in an urban area of Malaga city (Figure 1). The first Monk Parakeets appeared in the area in 1995, and since then, the population was regularly censused.

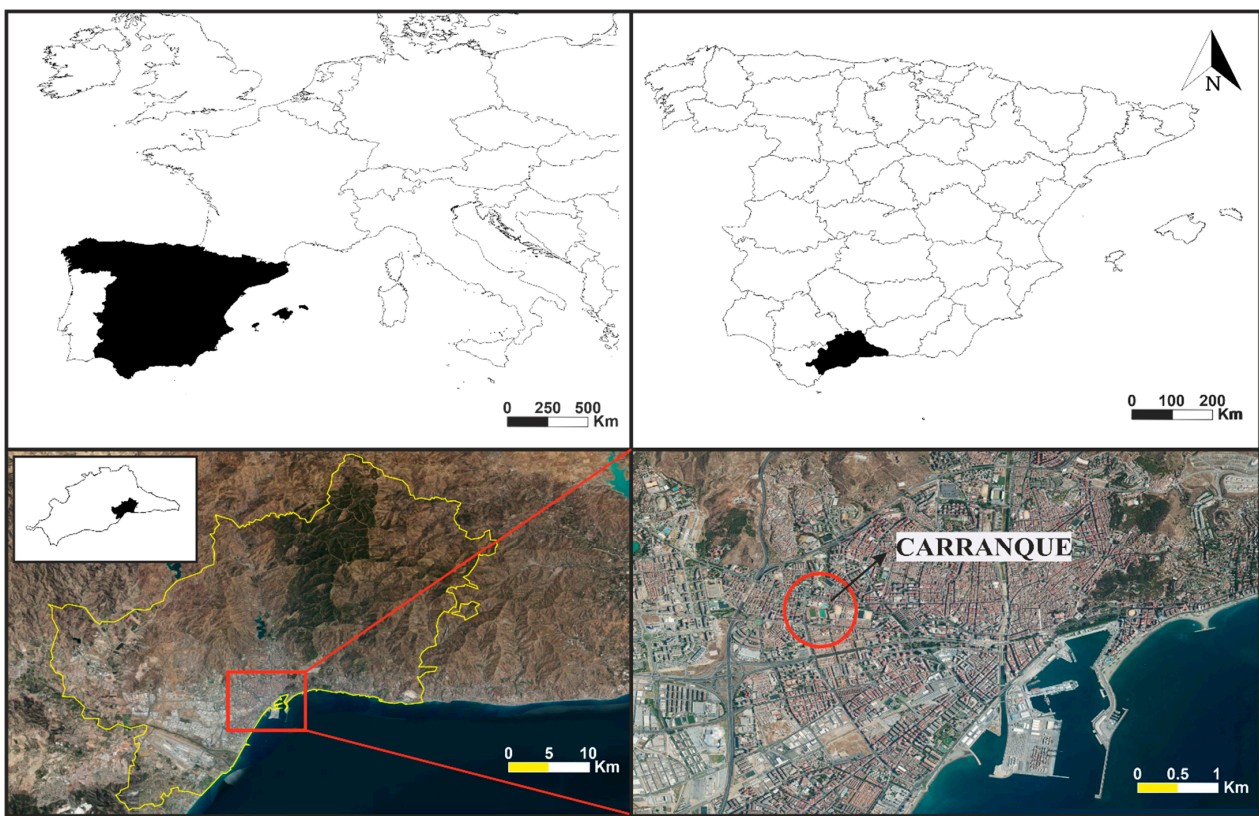

**Figure 1.** Location of the study area. The upper images show the province of Malaga, standing out within the Iberian Peninsula, and the lower images show the municipality of Malaga in the provincial context and Ciudad Deportiva de Carranque (36°43′ N, 4°26′ W) within the municipality of Malaga.

The vegetation of the area included eucalyptus trees (*Eucalyptus* spp., 9 individuals), cypresses (*Cupressus sempervirens*, 7 individuals), ficus (*Ficus macrophylla*, 2 individuals) and palm trees (*Phoenix canariensis*, 27 individuals). The landscaped areas were mainly composed of grass and a large variety of plants. With the exception of ficus, all of the trees mentioned are used as nest substrates by Monk Parakeets in the study area. In addition, Monk Parakeets can build their nests on electric utility structures, of which six were located in the study area.

### 2.2. The Species

One of the particular features of the Monk Parakeet is that their nests are constructed of tightly intertwined twigs and sticks [36], and are often comprised of several independent compartments (chambers) with separate entrances. Two types of nests can be distinguished: single-chamber nests (small housing for a single pair with only one chamber) and communal nests (large-size housing with multiple chambers accommodating several pairs or groups). The nests are used throughout the year for both breeding (during the breeding season) and roosting (all year round), and they are constantly repaired and can be reused [37,38]. In Spain, as well as in other areas where the species has become established, Monk Parakeets still occur mainly in urban and suburban areas (especially in gardens and

parks) [39], and nests have been mostly found in palms (*Phoenix* spp.), cedars (*Cedrus* spp.), pines (*Pinus halepensis*), planes (*Platanus* spp.), eucalyptus trees (*Eucalyptus* spp.), cypresses (*Cupressus* spp.), poplars (*Populus* spp.), elms (*Ulmus* spp.) and artificial structures such as utility poles and light towers [13,33].

*2.3. Data Collection and Analysis*

To assess population status, we counted the number of Monk Parakeets, the number of nests and the number of chambers per nest during the years 1992, 1994, 2002, 2007, 2012, 2013, 2015, 2016, 2017, 2018 and 2019. We used the same methodology for detecting nests in each survey. Monk Parakeet's nests are conspicuous and easily detectable because birds use them throughout the year. To get a more accurate estimate of the total number of nests and chambers, we conducted our counts before the breeding season, which starts around April in the Mediterranean Basin [40]. In 1992, the species was not present yet in Ciudad Deportiva de Carranque, and in 1995, only one parakeet was observed in the area, but no nests. Starting in 2002, we looked for nests throughout the study area and also in the surroundings (within a radius of approximately 500 m) to monitor the dispersal process and identify their substrates.

We conducted population censuses three times a year, also before the breeding season. One or two people participated and were placed in a specific location where they had a full view of the study area. To estimate the number of Monk Parakeets in Carranque, we counted them in an open grassy area in the middle of the study area because Monk Parakeets tended to congregate there before roosting. Therefore, all censuses were conducted at dusk. As the number of individuals began to increase over the study period, we took pictures of the groups to obtain more accurate estimates.

Following the methodology proposed by Van Bael and Pruett-Jones [31], we used the number of Monk Parakeets detected in the period 1995–2019 to estimate population growth trends. To obtain the rate of population growth (*r*), we fitted the census data to the model using the equation $N_{t+1} = N_t \times e^{rt}$, where $N_{t+1}$ is the population size at time t + 1, $N_t$ is the population at time t, *e* is the natural logarithm base, *r* is the intrinsic rate of population growth and *t* is the time interval. To check if the number of nests and chambers were correlated, we performed a Pearson correlation with IBM SPSS Statistics (Version 25) software.

**3. Results**

The population grew rapidly from 1995 until 2015 and stabilized from 2016–2018, reaching its highest point in 2019 (Figure 2). The maximum growth rate was 0.314 and occurred during the years 2002 and 2007, while it was in 2016 when the population stabilized and even minimally declined (*r* = −0.043). The average growth rate was 0.191 (model $R^2$ = 0.934). The number of nests and chambers also grew rapidly until 2015, when the number stabilized and then decreased slightly at the end of the study period. During the last two censuses, there was a slight decrease in the number of nests and chambers, even as the number of Monk Parakeets slightly increased (Figure 2). The number of nests was significantly correlated with the number of chambers (Pearson correlation test 0.995, *p* < 0.01). During the study period, the maximum Monk Parakeet population of 100 individuals was reached in 2019; the largest number of nests (32) and chambers (59) was reached in 2017.

In 2015, once the maximum population size was reached, new nests appeared in the near vicinity of our study area. While the number of nests stabilised in the Ciudad Deportiva de Carranque sports complex, three new nests were found in 2019 at distances of 20, 50 and 100 m, outside our study area.

Most of the Monk Parakeet's nests in the study area were found in palms (Phoenix canariensis); a few were placed in electric utility structures, and two were placed in a eucalyptus tree (Table 1).

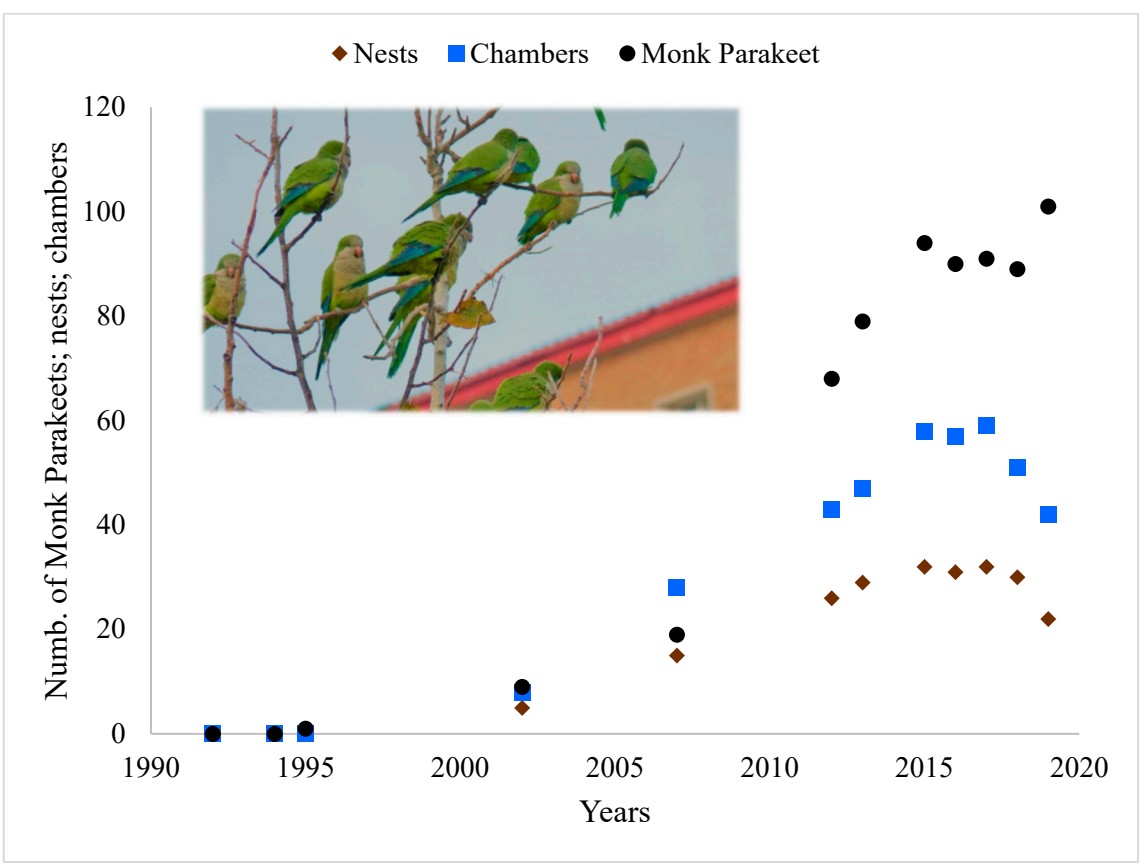

**Figure 2.** Estimated Monk Parakeet population size from 1992 to 2019, measured in terms of the number of observed parakeets, nests and nest chambers.

**Table 1.** Nesting substrates used by the Monk Parakeet in the study area in 2017.

| Nesting Substrate | Available | Used | % Used | Number of Nests | % Nests | Number of Chambers | % Chambers |
|---|---|---|---|---|---|---|---|
| *Phoenix canariensis* | 27 | 24 | 88.88 | 23 | 71.875 | 33 | 55.93 |
| Electric utility structures | 6 | 5 | 83.33 | 7 | 21.875 | 22 | 37.29 |
| *Eucalyptus* spp. | 9 | 1 | 11.11 | 2 | 6.25 | 4 | 6.78 |
| TOTAL | 42 | 30 | | 32 | 100 | 59 | 100 |

## 4. Discussion

The Monk Parakeet is one of the most widely distributed and successful introduced parrots in the world [41]. The number of cities where it is creating problems and is starting to require control efforts is growing as the species continues increasing in numbers and geographical range [13,42,43]. In Spain, it continues to increase its range as predicted by the distribution models proposed by Muñoz and Real [12] and Muñoz [30], colonizing new areas via urban settlements. Urbanization has created a number of new ecological niches, which, after remaining empty for some time, are increasingly being colonized by both native and invasive birds [26,44]. For this reason, detailed studies are needed to learn more about the growth and dispersal dynamics of the species to mitigate potential environmental impacts and conflicts with humans. Aside from ours, no studies have monitored Monk Parakeet populations at a local level over the long term as an invasive species. In our study, we obtained population counts starting from the time of introduction to the study area until the population size appeared to have stabilized, possibly indicating that it had reached carrying capacity.

According to our results, our study population appeared to reach a stable size, possibly indicating the habitat's carrying capacity. This result may be of great interest to estimate the number of Monk Parakeets that a given area can support and also to better predict the future occupation of neighbouring areas. In this respect, it would be interesting to deepen our knowledge of the species' diet as well as its home range because the resources that limit the population growth of this species are not known. These limiting resources could include nesting substrate and/or food availability. In our study case, once the population reached the maximum sustainable level of population size in 2015, other nests began to appear in the vicinity, outside the sports facilities. In the following years, the number of nests has been increasing in the surroundings but has remained constant in our study area. Although we do not precisely know the origin of the birds that have formed these new colonies because we have neither marked individuals nor carried out genetic studies, we think that these specimens came from the neighbouring and already saturated population. Our study population was isolated from other Malaga populations at the time it appeared (approximately 6 km away) and, to date, remains relatively isolated. At the time when the population reached its peak, it could have begun to act as a source of individuals, as was found for a Monk Parakeet population in Barcelona (Spain) using molecular genetic data [45].

The most common nesting substrate in Spain is palm trees of the genus Phoenix [13,46]. This fact coincides with our observations where most of the Monk Parakeet's nests were found in palm trees of the species *Phoenix canariensis*. This may be due to the high availability of palms in our study area. Eucalyptus trees are a preferred nesting substrate in their native distribution [47], and a range expansion of the species in Argentina was facilitated by the planting of these trees in grassland areas [48]. Similarly, eucalyptus trees were the main substrate in Andalusia at the end of the 20th century [24], but they have become a secondary substrate more recently, with palm trees now being preferred [13]. The second main nesting substrate we have found in our study area is electric utility structures. The use of these structures has also been observed in the United States, where it causes costly damage to electrical infrastructure due to short-circuiting or overheating, resulting in power outages, fires and electrical service disruption [49–51]. Currently, every year the nests in our study area are removed from utility structures to avoid fires during summertime. If the parakeets continue to build nests on electric utility structures, the economic cost and damages associated with the removal of nests will continue to grow.

Our study area is an example that suggests that range expansion in Monk Parakeets may occur by neighbourhood diffusion, occupying favourable adjacent areas. The general trend of the species in Spain is to spread, occupying practically all city gardens and parks [43,52], preferably those close to areas that are already occupied. At the city level, once favourable urban areas (mainly parks and gardens) are occupied, and the species reaches high density, Monk Parakeets are likely to begin colonizing suburban and rural areas, as is being observed at present in Malaga province (pers. obs.). In this area, the potential for some negative impacts exists. If the parakeet population were to spread into the rural areas, it could cause some of the negative economic and environmental effects that have been observed in other areas [43,52]. For this reason, it is necessary to know, in detail, the home range size, the dispersal process and population change over time, bearing in mind that it is easier and less costly to prevent the settlement of new individuals than to try to eliminate a parrot population once it is established.

**Author Contributions:** Conceptualization, A.-R.M.; Data Collection, both authors; Writing, both authors. All authors have read and agreed to the published version of the manuscript.

**Funding:** This research received no external funding.

**Institutional Review Board Statement:** Not applicable.

**Informed Consent Statement:** Not applicable.

**Data Availability Statement:** Not applicable.

**Acknowledgments:** We appreciate the collaboration of the Carranque Sports Facility for allowing access to the area to conduct regular censuses and nest and chambers counts. We also appreciate the comments and suggestions of two anonymous reviewers, which have improved the initial version of the manuscript.

**Conflicts of Interest:** The authors declare no conflict of interest.

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
