# Peer review of "A Local Approach to Better Understand the Spread and Population Growth of the Monk Parakeet as an Invasive Species"

_2673-6004, doi:10.3390/birds3030018_

Round 1
Reviewer 1 Report
This manuscript presents data on annual population and nest counts for a small introduced population of Monk Parakeets in Spain. Spanning nearly two decades, the census data provides a nice description of population dynamics for the population, starting at the time when the species was first detected in the study area.
General comments
- For a recent compilation of information on introduced parrots, the authors should have a look at “Naturalized Parrots of the World” edited by S. Pruett-Jones.
- I would recommend defining “local” and “regional” (as they apply to your study) in the Introduction. I would also suggest introducing and defining the concept of “carrying capacity” in the Introduction. Although the concept is straightforward in theory, its application to wild populations can be tricky since it typically assumes no interactions between species. You should discuss how one can detect/determine whether a population has reached carrying capacity; the observation that a population has stopped growing does not necessarily mean that it has reached carrying capacity. For example, strong hunting pressure/poaching, or disease, can depress population size below carrying capacity. On the other hand, a population can grow (even exponentially) beyond carrying capacity, only to crash later. In the case of Monk Parakeets, you could conceivably have a cultural shift in nest substrate use result in population growth: the initial colonists could select trees for nesting, and only later begin nesting on utility structures, resulting in an (apparent) increase in available nest sites, leading to increased breeding and population growth. This sort of nesting shift, correlated with population growth (note: cause/effect not demonstrated), has been documented for Monk Parakeets in Belgium (Weiserbs & Paquet. 2016. Recensement de la conure veuve Myiopsitta monachus a Bruxelles en 2016. Aves 53:19-28). I think it’s fine to frame your study in terms of the carrying capacity concept, but as currently written this is done a bit too simplistically. A paper that might be useful: Dhont (1988) Carrying capacity – a confusing concept. Acta Oecologica 9(4):337-346. (no doubt there are others that are relevant here; this is just a suggestion as a starting point)
- I think the Discussion could be condensed a bit. There is some repetition of the idea that the study population expanded to a certain point, and then may have served as a source of individuals that (may have, or apparently) colonized neighboring areas.
- It would be useful to provide information on the distribution of Monk Parakeets in Spain, especially in relation to the study area. How isolated is the study population relative to other areas where the species is found? How does the population size and timing of introduction compare with other Monk Parakeet populations in Spain?
- Recognizing that English is likely a second language for the authors, I have provided extensive comments to improve grammar and language usage.
Detailed comments:
l.10 “in new ecosystems” … “may be difficult and costly”
l.14 “an declined slightly”
l.15 “a limit to the number”
l.16 “favors the dispersal of individuals to”
l.17 “areas. Once they’ve reached maximum size, populations serve as sources… for the colonization”
l.18 “adjacent areas” … “Monk Parakeets show a preference for building their nests in palm”
l.19 “and on electric utility structures, and this preference should be”
l.20 “evolution” has a very specific meaning in biology, and is not what you mean here. Perhaps reword: “dispersal process and population change over time”
l.22-23 The first sentence of the abstract is awkward and should be re-written. Mention of “a great debate” implies that there is a What do you mean by “which get to the point”?
l.25 “in some large cities”… “are growing quickly”
l.26 “continues to expand”
l.26-27 “a population that has been closely monitored”
l.27 “growth and change on a local scale” (see comment for l.20)
l.28 “occurred in 2002”
l.29 “and declined slightly”
l.31 “study area”
l.32-33 “with populations that have reached maximum size serving as sources of individuals dispersing to suitable areas. In our…”
l.34 “shows a strong preference for building its nests in palm trees”
l.35 “Our study’s local focus in studying the population dynamics of an invasive species allows us to explain..."
l.36 “management strategies”
l.41-42 “as a direct or indirect result”
l.44 “from the area where they were initially introduced.”
l.44-45 “Invasive species also pose a significant threat to the species, ecosystem, and/or economy of their new range.
l.45-47 This sentence could be condensed or simplified, e.g., “Most introduced species do not become invasive.”
l.48 delete “up to now”
l.49 “have been recorded” or “have been documented”…”but less than 10%”
l.50 delete “themselves”… “a rise in the legal and illegal pet trade”
l.51 “across international borders”
l.52 “among the most”
l.58-59 Delete “Once barriers….surpassed”
l.60 “successful establishment in new habitats.” This wording is important, because Monk Parakeets have not become invasive at every location where they have been introduced.
l.60 “loud and harsh call”
l.65 “builds its nests rather than using cavities for breeding. This facilitates…”
l.66 “compete with other species for available cavities for nesting.”
l.68 “were observed”
l.70 “including Madrid and Malaga”
l.70-71 “the species’ distribution and population size continues to grow, and has”
l.72-73 “Like many…species, after becoming established, the Monk Parakeet population has grown exponentially since the first breeding individuals were detected.”
l.74 “Wherever the species has been studied…”sounds like a global statement, but exponential population growth has not occurred at all places where MPs have been introduced. Are you only referring to MP populations in Spain?
l.76 Just because a population stops growing does not necessarily indicate that it has reached carrying capacity. (see General Comments)
l.74-82 Since you are making a distinction between “regional” and “local” that seem to be key to your argument, it would be good to define what you mean when you use each of those terms.
l.76 “saturated” is not the correct term here; I think you mean that the parakeets have reached high densities in a particular location?
l.85-86 “The study area was located in southern Spain, and comprised a sports facility…”
l.87 “The first Monk Parakeets appeared…”
Fig.1 I like the “zooming in” format of this figure. Just one missing link: it is not clear how the 2nd and 3rd panels are linked (the darkened area in panel2 is not the same shape as the area outlined in yellow in panel3) The city of Malaga should be indicated on panel2, and in the figure caption, you can state that in panel3 the city limits are outlined in yellow.
l.96 “With the exception of ficus, all of the trees mentioned are used as nest substrates by Monk Parakeets in the study area.” (this is a suggested wording, but should be edited for accuracy; it’s best to avoid saying “normally built its nests”… what is “normal” will depend on the particular habitat)
l.103 “have independent compartments (chambers) with separate entrances.”
l.104 I would suggest using “single-chamber nests” rather than “simple nests,” since the latter implies “simplicity” (and that is not what you intend to communicate here)
l.105 “large size with multiple chambers accommodating several pairs or groups.” Since the parakeets use their nests year-round by both breeding and non-breeding individuals, it’s important to word this sentence in such a way that you avoid implying that nests are only occupied by breeding pairs.
l.105 I am not sure what you mean by “complex structured housing.” Monk Parakeet nests are built via accretion, by adding chambers onto an existing structure, so that the communal nest gradually increases in size.
l.109 “other areas where the species has become established” It is important to be careful about your use of “invade/invasive” because this species is not necessarily invasive at every location where it has been introduced.
l.113 “such as utility poles and light towers”
l.116 see comment for l.20 “assess population status we counted…”
l.120-121 delete “behavioral habit…detection” (repetitive)
l.121 “more accurate estimate of”
l.122 “we conducted our counts before…”
l.124 “Starting in 2002 we…”
l.127 “We also conducted population censuses before…”
l.128 Please provide a better description of how censuses were conducted. Explain briefly what is meant by “counting them simultaneously and taking pictures to[of] the groups”. At what time of day were the counts made? Were the counts made on a single day? If so, how many people participated, and how were they positioned? (or did they move around?)
l.131 “to a model” Please cite a reference for this methodology.
l.138 “Population counts showed that…”
l.140 “with an average rate”
l.140-142 According to what is shown in Fig.2, I would suggest switching the sentence order and rewording: “….(Figure 2). The population grew rapidly from 1995 until 2015, declined slightly from 2016-2018 (r = -0.043), and then increased to its highest point in 2019. The maximum growth rate…”
l.143 “when the number stabilized and then decreased slightly at the end of the study period.”
l.145 “chambers, even as the number of…”
l.147-148 “The study area has supported…” No need to repeat the size of the area, since that information is provided in the previous section.
l.154 “in palms (Phoenix canariensis); a few were placed in electric utility structures, and two were place in a eucalyptus tree (Table 1).”
L161 “and is starting to require control efforts is growing”
l.162-163 “In Spain, it continues to increase its range… predicted by the distribution model proposed by Muñoz [28]”
l.166 “Aside from ours, there are no studies that have monitored… over the long term”
l.167 “In our study we obtained population counts starting from the time of introduction to the study area.”
l.168 It seems like it would be difficult to justify the statement that “the population reached carrying capacity.” Although the population size plateaued after a period of exponential growth, the final census count also suggests that the population could possibly be entering another growth phase. At the very least I would suggest that you temper your wording, e.g. “until the population size stabilized, possibly indicating that it had reached carrying capacity.”
l.169 “The population growth trajectory and…” or “The population dynamics and…”
l.170 see comment for l.20
l.172-174 This sentence is awkwardly written. A possible re-write: “The carrying capacity of an environment is the maximum population size of a species that can be maintained without negative effects on the habitat.”
l.174-175 “This maximum population size depends on…” (the definition does not involve the concept of a threshold, so that word shouldn’t be used here)
l.177 “saturation” is not used properly here; it would be more correct to discuss your results in terms of population density, which may have reached a maximum sustainable level (but see General Comment re. carrying capacity)
l.180 The number of monk parakeets that can live sustainably in an area will depend only in part on the size of the area. It is important to point out that the resource(s) that limit population growth in this species & habitat are not known, and could include nesting substrate availability and/or food availability.
l.181 see comment for l.177
l.182 If you have data/observations on this, please include in the Results section
l.182 “These were likely…” You cannot be sure without evidence (e.g. banded/marked individuals, or genetic data).
l.184 “At that point, the population would have begun to act as a source of individuals…”
l.185 “the study area into neighboring areas”
l.185 Was this genetic study conducted using samples from your study population? If so, then you should state this explicitly. If not, then reword, e.g. “as was found for a Monk Parakeet population in XX using…”
l.187 “to our observations, Monk Parakeets in our study area showed a preference for nesting in palms” NOTE: I do not think your data actually show a preference (except, perhaps, an avoidance of eucalyptus trees). To demonstrate a preference, you need to do a Chi-square test to compare the use of particular nest substrates vs. the availability of those substrates in your study area. Eyeballing Table 1, it looks like the parakeets are not showing a preference; most of their nests are in palms simply because more palms are available.
L.188 It would be good to check these references to determine whether their data/analyses demonstrate nest substrate preferences, or simply frequencies of use for different substrates.
l.191-192 “Eucalyptus trees are a preferred nesting substrate in their native distribution…” And you might point out a range expansion in Argentina was facilitated by the planting of eucalyptus trees in otherwise treeless grassland areas (see Bucher & Aramburu. 2014. J. Biogeography 41:1160-1170)
l.196 “causing costly damage to electrical infrastructure in the United States”
l.198 Please clarify what you mean by “these nests”… the ones in the United States, or the ones in your study area?
l.199 “If the parakeets continue to build nests on electric…” For some reason that defies logic, in English it’s customary to say that nests are built “in trees” but “on utility poles/structures.”
l.201 “continue to grow”
l.202 “Monk Parakeets spread”
l.202-203 Although there’s nothing wrong with poetic language, I would recommend sticking with a more technical style, and avoid phrases such as “spreads locally like an oil stain”
l.203 “areas that, once saturated, may act as sources for adjacent areas.” Or “new populations in adjacent areas” (the areas themselves are not new) NOTE: Although you mention some anecdotal observations of nests appearing in neighboring areas, this ms does not present data showing that once an area houses a maximum density of individuals, those individuals then disperse to adjacent areas. Be careful to limit your conclusions to statements that are supported by your data.
l.204 “level, once favorable urban areas…”
l.205 “occupied and the species reaches carrying capacity” (see General Comment re. this concept)
l.205 “Monk Parakeets are likely to begin colonizing” or “may start to colonize” (I would avoid making such a strong prediction, e.g. saying “will start to colonize”)
l.206-207 “as has been observed at present in our study area, where tropical fruit crops such as mango and avocado predominate.” Please cite a reference to back up this assertion.
l.208 “If the parakeet population were to spread into the rural areas of surrounding provinces, it could cause some of the negative economic and environmental effects that have been observed in other areas”
l.210 Please check your references here. Although you imply that these references show that Monk Parakeets have caused economic and environmental damages in other areas, that is not the case. E.g., [48] concludes that Monk Parakeet crop damage in Argentina appears to be overestimated; [51] presents data for Ring-necked Parakeets (not Monk Parakeets); and [49], [50] are conceptual papers that do not present data on Monk Parakeets.
L.211 see comment for l.20
l.211-212 I would avoid informal phrases like “prevention is always better than the cure,” and stick to a technical style.
Reviewer 2 Report
Comments are included in the pdf file.

Round 2
Reviewer 1 Report
This manuscript presents data on the growth of a Monk Parakeet population in Malaga, Spain. These data are useful for understanding the population dynamics of this introduced species, which has also been studied in other areas of the country. The writing/grammar is improved compared with the previous version, but some further editing is necessary (see Detailed Comments below).
General Comment:
- In the introduction (lines 91-94) the authors state that the population growth was monitored “at a local level” (presumably the Ciudad Deportiva Carranque) and that they also sought to determine when that population began to act as a source. This would require monitoring of surrounding areas in order to detect new breeding colonies, however such monitoring is not described in the Methods section.
- I would encourage the authors to be careful in presenting conclusions, and to stick close to what their data show. It’s ok to speculate/hypothesize a bit, speculative statements should not be presented as conclusions.
Detailed Comments:
l.11 “a Monk Parakeet population”
l.17 “suggest that there may be a limit to the number”
l.18 “favor the dispersal”
l.18-19 The sentence “Once they…. adjacent areas” is written a statement of fact, but it is actually a hypothesis. You do not actually know that this is the case (e.g., the population may well serve a source population before it reaches a “maximum size.” Refrain from overstating your results.
l.19 “populations may serve as sources”
l.22-23 I would refrain from using informal expressions like “prevention is always better than [the] cure”
l.24-25 “management agencies”
l.28 “we focused”
l.30 Your manuscript really addresses population size, not density.
l.39 I would argue that your data do not “explain the increased range of the species on a larger scale” Your data contribute to the body of information that would be necessary for such an explanation, but this statement over-sells your results.
l.49 “may/can also pose a significant threat”
l.61-62 I would suggest moving the sentence “Some characteristics…” to follow “…Italy and Belgium).” And edit the sentence that follows to “An important characteristic…” Another aspect of the species that facilitates establishment in new habitats is that they are generalist foragers, and feed on a variety of food types (cite ref.).
l.80 What do you mean by “local level”? In the previous sentence you refer to Malaga and Spain… would Malaga not be considered “local”? This needs to be clarified (see comments for the previous ms version)
l.92-94 “starting from the time at which the species was first recorded in the area, to determine the area’s carrying capacity, and whether the population has become a source of new breeding colonies in adjacent areas.”
l.99-100 “a 58.360 m2 sports facility (Ciudad Deportiva Carranque) in an urban…”
l.103 “Ciudad Deportiva Carranque”
l.105-107 I would suggest using “individuals” rather than “specimens” here; usually the latter term refers to sample (collected) organisms
l.110-111 “of which six were located in the study area”
l.113-115 “their nests are constructed of tightly intertwined twigs and sticks, and often comprise of several independent…” This rewording is more correct, because (as you state in the following sentence), not all structures have multiple chambers, and are thus not all “communal structures.”
l.118 “throughout the year, for both breeding…”
l.136 What exactly do you mean by “and its surroundings”? How large was the censused area, and/or how far did your monitoring extend beyond the boundaries of the sports facility? In line 99 you define the study area as being the sports facility… but apparently you monitored additional areas. The monitored area should be defined clearly and more precisely.
l.142 “tended to congregate there”
l.156-157 As written, you state that in one year (2016) the population both stabilized and declined. This doesn’t really make sense, since both population stability and decline are defined by changes in population size over time (i.e., over several years).
l.168 “evolution” has a very specific meaning in biology, and is not what you mean here. Possible re-wording: “Estimated Monk Parakeet population size in... measured in terms of the number of observed parakeets, nests, and nest chambers.”
Fig.2 It would be nice to provide lines for each of the data types (either line/model fits, or just lines that join adjacent data points)
l.185 “no studies that have monitored populations at a local level over the long term” is a very vague statement… do you mean Monk Parakeet populations? Or populations of introduced species? How do you define “local” and “long-term”? I recommend making your language/statements much more specific and clear.
l.187 “appeared to have stabilized”
l.188-190 I would avoid making vague, sweeping statements like this one, and focus on clear conclusions that can be drawn from your data.
l.191-192 “Our study population appeared to reach a stable size, possibly indicating the habitat’s carrying capacity” Note: your results were reported in terms of population size, not density, so you should avoid talking about density here.
l.193-194 “This result…” The number of parakeets that can be supported in an area is critically dependent on the resources (e.g., food, nesting substrate) in the area, and these resources can be distributed quite heterogeneously across the landscape. The population size alone is not sufficient to make the estimate that you are suggesting. For your data to be useful in the way that you are suggesting, you would need to present more data on resource availability (and also information on how the parakeets access the resources; e.g. how far they travel to forage). If they travel far to forage, then their home range (and, therefore, the area required to maintain your study population) may extend well beyond the study area.
L.198-200 This information on new nests should be reported in the “Results” section, and an explanation on how these data were gathered (systematically in censuses? Opportunistically?) should be presented in the Methods section.
l.204 Perhaps mention how far away the nearest known population is.
l.204 What makes you suspect that the initial colonists came from a neighboring population? Couldn’t the initial colonists have been escaped/released pets?
l.206 “and to date remains relatively isolated.”
l.206-208 The sentence “At that point…” is a bit confusing as written. At what point? Which population are you referring to—the study population, or a hypothetical source population?
l.214-215 “Similarly, Eucalyptus trees were the main nesting substrate… century, but they have become a secondary substrate more recently, with palm trees now being preferred.”
l.218 “is electric utility structures.”
l.218-219 “The use of these structures has also been observed in the United States, where it causes costly damage to electrical infrastructure due to short-circuiting…”
l.221 Are the nests only removed from utility structures, or from all nesting sites?
l.225 This study does not present data showing that range expansion occurred by neighborhood diffusion. The data presented in the ms show the pattern of population growth of the study population, and some anecdotal observations of new nests in the general area are presented, but these observations seem fairly limited. It is important to be careful about drawing conclusions from the data at hand, and not presenting speculations/hypotheses as fact.
L.226-228 The statement “The general trend….” needs a citation, or data, to back it up.
l.231 When you say “study area” here, it seems that you are referring to a much larger area than the “study area” defined in the Methods. Are the rural areas that you mention close to the study area?
l.234-235 It would also be necessary to know the parakeets’ home range size and have information on the distribution of important resources (especially food and nest substrate).
Reviewer 2 Report
I think that the manuscript can be accepted in the present form. The authors did some changes and improvements and they have improved the quality of the manuscript notably. Thank you very much for your confidence.
Author Response
Thank you very much for your help and suggestions for improvement.
Round 3
Reviewer 1 Report
I have reviewed the changes made in response to earlier comments, and just have a few minor suggestions regarding the revisions. Upon reviewing a suggested change to Fig.2, my attention was drawn to the log-transformation of some of the data (something that I had not previously focused on), and so I ended up having a new item to fix (see below)
l.12 The abstract refers to population density, but in fact the manuscript reports data on population size, not density. I had flagged this for correction at l.30 (and this correction was made by the authors), but it should also be corrected in the abstract.
l.29 “The maximum sustainable population size…”
l.39 “may allow us to understand…” (If you say “allows us to understand” then you have to demonstrate this in the manuscript; as currently written, you do not.)
l.61-62 I think the authors have mis-interpreted my comment regarding the monk parakeet’s generalist diet. My point is that because they feed on a wide variety of food types, they are quite adaptable to new environments where the food resources may be quite different from their native habitats. This has been suggested in the literature, e.g. South & Pruitt-Jones 2000 (Condor 102:848-854)
l.116 There was a typo in the detailed comment. Should be “comprised”
l.166-167 “During the study period, the maximum Monk Parakeet population of 100 individuals was reached in 2019; the largest number of nests (32) and chambers (59) was reached in 2017.” The sentence “These were the maximum…” can be deleted, since it repeats what was stated in the previous sentence.
Fig.2 I had not earlier focused on the fact that the parakeets and nests/chambers are plotted according to two different Y-axes. I am not sure why nest & chamber counts are log-transformed, since the raw counts for nests/chambers fall within a range that could perfectly well be plotted without a log-transformation, just as the numbers of parakeets are plotted. I would strongly recommend plotting the raw counts for all three data types, If counts for one of the data types were orders of magnitude different from the others, then it would make sense to log-transform the data, but that is not the case here. Re-plotting in this way would also reduce the congestion where some of the parakeet and nest points overlap (and could allow for a line/model fit). I apologize for not noticing this log-transform issue earlier!
l.198 “that a given area can support”
l.200 “of the species’ diet as well as its home range, because the…”
l.201 “of this species are not known.” (because it doesn’t make sense to say that “the resources that limit the population growth of its habitat”… which is what the sentence implies, as written).
l.211-212 “it could have begun to act…”
l.213 “using molecular genetic data” or “using molecular genetic analyses”
l.225 I would recommend further toning down your conclusion: “that suggests that range expansion…”
l.235 “reaches high density”
note: an accent mark for the second “a” in “Malaga” used inconsistently
